

# Meniscus injury prediction model based on metric learning

Yu Wang[1,2], Yiwei Liang[3], Guangjun Wang[1,2,4], Tao Wang[1], Shu Xu[1,2], Xianjun Yang[1], Yining Sun[1] and Zenghui Ding[1,5]

[1] Institute of Intelligent Machines, Hefei Institutes of Physical Science, Hefei, Anhui, China
[2] University of Science and Technology of China, Hefei, Anhui, China
[3] DukeKunshanUniversity, Kunshan, Jiangsu, China
[4] Anqing Normal University, Anqing, Anhui, China
[5] Department of Mathematics and Computer Science, Tongling University, tongling, Anhui, China

## ABSTRACT

A meniscus injury is a prevalent condition affecting the knee joint. The construction of a subjective prediction model for meniscus injury represents a potentially invaluable diagnostic tool for physicians. Nevertheless, given the variability of pathological manifestations among individual patients, machine learning-based models may produce errors when attempting to predict specific medical records. In order to mitigate this issue, the present study suggests the incorporation of metric learning within the machine learning (ML) modelling process, with the aim of reducing the intra-class spacing of comparable samples and thereby enhancing the classification accuracy of individual medical records. This work has not yet been attempted in the field of knee joint prediction. The findings demonstrate that the adoption of metric learning produces better optimal outcomes. Compared to machine learning baseline models, F1 was increased by 2%.

Corresponding author
Zenghui Ding, dingzenghui@iim.ac.cn

## INTRODUCTION

Knee joint disease represents a prevalent pathological condition. It encompasses a range of disorders, such as knee osteoarthritis (KOA), meniscal injuries, patellofemoral disorders, and cruciate ligament disorders, which occur commonly in clinical practice. Notably, the structural complexity of the knee joint engenders overlapping symptomatology among these disorders, posing significant diagnostic challenges to clinicians. In this regard, effective screening and prediction of meniscal diseases through the identification of subjective symptoms and disease history may serve as a valuable diagnostic tool, providing physicians with relevant diagnostic criteria and patients with convenient initial screening methods. This approach holds considerable research significance in the quest for optimal diagnosis and management of knee joint disease.

In recent years, artificial intelligence (AI) technology has been developed for increasing the accuracy and efficiency of medical diagnoses (*Currie et al., 2019*; *Shen, Wu & Suk, 2017*). AI has been applied in various domains, including the development of fracture and

dislocation diagnoses (*van Spanning et al., 2022*; *Beaugerie, Rahier & Kirchgesner, 2020*). *Chan et al. (2021)* have employed machine learning algorithms to forecast the prognosis of knee osteoarthritis through the analysis of radiography and symptom data, encompassing anthropological details, knee trauma history, metabolic syndrome, and lifestyle data. In a separate investigation, *Kwon et al. (2020)* utilized machine learning algorithms to gauge the severity of knee osteoarthritis by integrating step features with the Western Ontario and McMaster Universities Arthritis Index (WOMAC) scale using linear regression. *Bacon et al. (2022)* have put forth a proposition that a sensory modality, like an image of the knee, can be utilized to categorize osteoarthritis (OA) into three distinct types, predicated on the number of fingers curled in the subjects' hands.

Owing to the distinctive nature of medical samples, significant inter-individual variations exist in medical histories. traditional machine learning applied to medical data modelling faces two main challenges:

(1) Inconsistent interpretability of feature selection: Traditional machine learning methods for feature selection may produce results that contradict practical medical knowledge, making them difficult to interpret in the medical context. The need for interpretable features often requires involvement from domain experts, such as healthcare professionals, which can be a time-consuming process (*Kaya & Bilge, 2019*). Paradoxically, while having more features might be advantageous for the modelling process in general, indiscriminate feature selection may not be the best choice for medical data modelling.

(2) Patient-specific variability: Medical data inherently exhibits patient-specific variations, and the presence of specific cases can significantly impact the classifier's performance. Improving the construction of classifier hyperplanes can contribute to enhancing prediction accuracy, given the distinctiveness of individual patient cases.

The reason for the above problems is that machine learning algorithms need to construct classification planes based on sample features. In machine learning, determining the classification plane (also known as the decision boundary or hyperplane) is a crucial step that involves effectively separating samples from different categories. The division of hyperplanes involves determining a boundary line and dividing samples of the same class into the same area as much as possible. Traditional machine learning techniques typically operate within a Euclidean space, where the unit measurementis fixed and cannot be adjusted, posing limitations, particularly in domains like medical diagnostics. In such fields, where datasets often consist of highly specific and smaller sample sizes, this leads to significant differences in the distribution of samples of the same class, with larger intra-class intervals and smaller inter class intervals. In a fixed Euclidean space, the distribution of these samples poses a challenge to the partitioning of hyperplanes.

In the field of computer related research, some scholars have proposed metric learning theory to improve the similarity between identical samples (*Xing et al., 2002*). Metric learning addresses these limitations by learning a transformation that maps data to a new feature space, thereby altering the underlying metric. That is, by learning the corresponding algorithm through metrics, learning the distribution patterns of samples, redefining the unit metric, and obtaining the spatial transformation matrix corresponding to the Euclidean space, all samples are transformed into a new feature space. This transformation

reconfigures the original feature distribution, with the objective of minimizing the distance between samples of the same class while maximizing the distance between samples of different classes. By optimizing these intra-class and inter-class distances, metric learning enhances the classifier's performance in the new feature space. This theoretical advancement allows for more effective classification, particularly in scenarios where traditional Euclidean metrics are inadequate.

This technique has been employed in multiple domains to mitigate sample imbalance (*Wang, Xin & Xu, 2021*), large feature noise (*Guo et al., 2023*), and substantial intra-class (*Yang et al., 2021*) pacing of samples. Although distance metric learning has been recognized as a pivotal technique in the areas of pattern recognition and information retrieval (*Liang et al., 2020*; *Wu et al., 2021*), its adoption in the realm of medical prediction remains limited.

In the context of medical data, this study introduces metric learning into the modeling process for predicting meniscal injuries. The application of metric learning aims to enhance the similarity of features within identical samples, followed by the utilization of machine learning classifiers for classification. The primary contributions are outlined as follows:

(1) Building upon prior research, this study preprocesses questionnaire data and physical examination information related to meniscal injuries, rendering the data more universally applicable.

(2) The integration of large margin nearest neighbor (LMNN), one of metric learning, into the conventional knee joint prediction model enhances prediction accuracy by minimizing intra-class distances for highly specific medical records. This work is progressive in knee joint prediction modeling.

(3) In continuation of the research of *Wang et al. (2021)*, this article reassesses the impact of metric learning on machine learning classifiers.

## RELATED WORK

The application of artificial intelligence technology in the field of medicine facilitates expedited comprehension of patients' medical conditions by healthcare professionals. In the current phase, the predominant utilization of machine learning and deep learning (DL) technologies in the domain of meniscal injuries primarily involves the processing of imaging data through computer vision. By analyzing imaging data obtained from magnetic resonance and computed tomography scans, predictions pertaining to the onset and recovery status of meniscal diseases can be made.

*Kunze et al. (2021)* conducted a comprehensive review, examining a total of 11 studies from PubMed, OVID/Medline, and Cochrane that employed artificial intelligence models for the diagnosis of knee joint diseases. Among these, five studies focused on the diagnosis of anterior cruciate ligament (ACL) tears, five on meniscal tears, and one study simultaneously investigated both conditions. These investigations predominantly utilized imaging data to study meniscal pathology. The area under the receiver operating characteristic curve (AUC) of the artificial intelligence models ranged from 0.847 to 0.910, with a predictive accuracy spanning from 75.0% to 90.0%. The collective findings suggest that while

artificial intelligence cannot replace expert judgment, its performance in the realm of disease prediction is notably promising.

*Chou et al. (2023)* employed a dataset comprising 811 knee joint magnetic resonance imaging (MRI) studies to detect the position of the meniscus using the Scaled-YOLOv4 model. Subsequently, they utilized the EfficientNet-B7 model architecture to assess the presence of meniscal tears. The scaled-YOLOv4 model demonstrated AUC values of 0.948 and 0.963 in the sagittal and coronal plane views, respectively. On the other hand, the EfficientNet-B7 model exhibited AUC values of 0.984 and 0.972 in the sagittal and coronal plane views, respectively. Clinical practitioners can acquire structured reports on meniscal ruptures, enabling faster interpretation of images and saving valuable time in image analysis.

*Qiu et al. (2021)* conducted a study involving 205 patients from a hospital, comprising 24,60 images. They utilized a meniscus detection method based on the fusion of convolutional neural network 1 (CNN1) and convolutional neural network 2 (CNN2), referred to as CNNf. Through evaluation criteria such as accuracy, sensitivity, specificity, receiver operating characteristic (ROC) curves, and overall injury incidence, the study compared the accuracy of magnetic resonance imaging and computed tomography (CT) images in artificial intelligence-assisted knee joint predictions. This comparison aimed to effectively enhance diagnostic accuracy and reduce misdiagnosis rates.

Machine learning is not limited to processing images, it can also make predictions about patients' medical conditions by leveraging additional information such as electronic health records (EHR). Electronic health records provides valuable information about healthcare processes and their variations, making it a suitable tool for identifying and predicting adverse patient safety events (*Imtiaz, Shah & ur Rehman, 2022*).

*Tiulpin & Saarakkala (2020)* developed a multimodal pipeline for generating output predictions of osteoarthritis progression. By combining convolutional neural networks with other features such as age, gender, body mass index (BMI), injury, surgery, WOMAC scores, and Kellgren-Lawrence (K-L) grade, the best-performing model achieved an AUC of 0.81. This approach, which integrates multiple sources of information, demonstrates higher accuracy in predicting Osteoarthritis progression compared to using images alone.

*Yoo et al. (2017)* utilized the K-means algorithm to establish a scoring system based on predictive factors such as gender, age, body mass index, education level, hypertension, moderate physical activity, and knee pain. They evaluated the disease status of 60 rheumatoid arthritis (RA) patients, achieving an AUC of 0.81.

*Lezcano-Valverde et al. (2017)* employed random forest prediction on data from the Hospital Clínico San Carlos Rheumatoid Arthritis Cohort (HCSC-RAC; training; 1,461 patients) and the Princesa University Hospital Early Arthritis Register Longitudinal Study (PEARL; validation; 280 patients) to predict mortality in rheumatoid arthritis patients. The time-dependent specificity and sensitivity in the validation cohort were found to be 0.79–0.80 and 0.43–0.48, respectively.

*Norgeot et al. (2019)* proposed a technique based on long short-term memory (LSTM) for diagnosing and predicting the disease progression of 820 rheumatoid arthritis patients across different hospitals. The study results suggest that building accurate models to predict

**Table 1  Related research results.**

| Reference | Year | Technique | Dataset size |
|---|---|---|---|
| *Norgeot et al.* | *2019* | LSTM | 820HRS |
| *Lezcano-Valverde et al.* | *2017* | RRF | 1741HRS |
| *Yoo et al.* | *2017* | Kmeans | 60HRS |
| *Tiulpin & Saarakkala* | *2020* | ResNet | |
| *Chou et al.* | *2023* | Scaled-YOLOv4 | 811MRI |
| *Qiu et al.* | *2021* | CNN | 202MRI |

complex disease outcomes using electronic health record data is feasible, and these models can be shared across hospitals with different patient populations. Table 1 summarizes machine and deep learning technologies used for diagnosing arthritis in chronological order.

In the field of computer science, metric learning has been used in machine learning and deep learning classification algorithms. A key innovation in metric learning is its ability to address several common challenges in machine learning. By learning distance functions from data, this technique can resolve many of the inherent issues faced by traditional classification methods. For instance, by learning the similarity measurement between data points, it helps the classifier more accurately divide the boundaries between different categories. Improve the performance of classifiers in dealing with overlapping or uneven distribution of categories, as demonstrated by *Wang, Xin & Xu (2021)*, who showed that a learned metric can balance data distribution and thus prevent bias in classification.

Additionally, metric learning can help classifiers better understand the similarity structures in the data space in order to help classifiers overcome challenges such as noise, missing data, or outliers. For example, *Guo et al. (2023)* believe that in real-world data sets, the existence of inputs is a common problem, but metric learning can overcome this problem to some extent and ensure model performance.

Furthermore, as a consequence of diminishing intra-class spacing and augmenting inter-class spacing, the classifier exhibits enhanced discrimination capability among samples across distinct categories, thereby elevating classification efficacy. *Yang et al. (2021)* underscored the pragmatic utility of this approach in navigating a substantial volume of intra-class variations, substantiating that classifiers endowed with metric learning-based strategies evince heightened robustness across diverse datasets.

The pioneering attributes of metric learning render it a potent instrument within the domain of computer science, empowering robust, versatile, and precise classification endeavors. Its capacity to glean insights from data and dynamically adjust to diverse contexts presents a distinctive advantage, endowing metric learning with applicability across a broad spectrum of intricate challenges encountered in computer science.

The integration of metric learning with various classification algorithms has become a widely adopted approach, with researchers exploring innovative ways to optimize feature extraction and improve classification accuracy. *Wang, Peng & De Baets (2022)* introduced a bifurcated strategy that combined metric learning for feature acquisition with

extreme learning for classification. This unique combination demonstrated outstanding performance on benchmark datasets for scene recognition, showcasing the potential of metric learning to transform traditional classification methods.

In another innovative approach, *Park, Hong & Kwon (2024)* developed novel dimensionality reduction techniques, such as TripletPCA and ContrastivePCA, integrated into an end-to-end metric learning loss function. This integration not only optimized the features extracted by deep neural networks (DNN) but also resulted in improved performance on standard benchmark datasets for complex image classification tasks, illustrating the power of metric learning in enhancing deep learning frameworks.

Distance metric learning is also frequently used in medical data processing, especially in tasks such as medical image classification and similarity matching. Its working principle is the same as in other fields. By customizing distance measures based on specific features of the data, distance metric learning aims to minimize intra-class variance while maximizing inter class differences. Through this approach, the sample distribution space is reconstructed to improve the recognition rate of atypical medical records, enhance the sensitivity and specificity of the model, and enhance the performance of the classifier.

One notable application of distance metric learning in medical research is exemplified by *Lee et al. (2023)*, who developed CaMeL-Net for predicting cancer grading in pathological images. By leveraging centroids of different categories and integrating a deep neural network as a classifier, CaMeL-Net optimizes feature similarity, resulting in superior performance across multiple cancer datasets. Similarly, *Ma et al. (2022)* proposed MELT, a Metric Embedding Learning Triple network tailored for genomic data analysis. Through data-driven metric learning, MELT enhances the similarity between identical samples, thereby improving classification accuracy in genomic research.

Moreover, *Merchan et al. (2023)* demonstrated the versatility of DML in spectral analysis by employing a deep metric learning framework. By assessing similarity between mass spectra, they optimized neural network feature extraction, leading to the creation of a classification model for spectral data from various tropical disease vectors. This innovative application underscores the capacity of DML to address complex real-world problems beyond traditional domains.

Distance metric learningL has great potential in predicting knee joint diseases. By customizing distance metrics to capture subtle features of knee joint data, distance metric learning can effectively minimize intra-class variation while enhancing inter class differentiation. This method not only improves the accuracy of classification, but also helps to gain a deeper understanding of potential patterns related to knee joint diseases, ultimately helping to develop more effective diagnostic and treatment strategies.

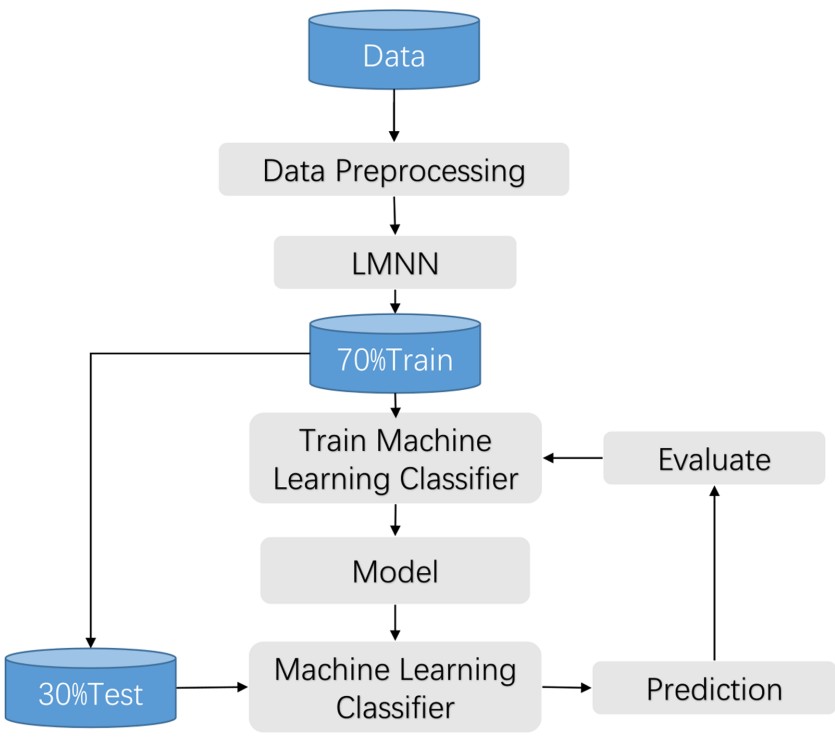

**Figure 1   Training process flow.**

## METHOD

### Overall framework

The clinical manifestations of knee pathologies are characterized by intricate and diverse symptoms, and meniscal lesions are strongly associated with other knee pathologies. Therefore, in this study, a synthesis of various data sources and literature, along with the expert opinions from the First Affiliated Hospital of University of Science and Technology of China (USTC), was conducted to collect information on various aspects related to knee disorders, including patients' gender, age, height, weight, knee work intensity, affected knee site, activity limitation, injury condition, popping condition, interlocking, instability, knee misalignment, patellar misalignment, stiffness, swelling condition, pain condition, pressure pain condition, and other related factors. Based on this information, a questionnaire was developed to assess the common symptoms associated with knee disorders. The questionnaire was reviewed and evaluated by the experts and subsequently refined to form a comprehensive tool for assessing the symptom performance of knee joint diseases. This work has been described in detail in the previous achievements of *Wang et al. (2021)*. The flow chart is shown in Fig. 1.

The large margin nearest neighbor algorithm constitutes a method employed within the domain of metric learning, aimed at discerning a linear transformation such that instances belonging to the same class exhibit reduced inter-instance distances, while concurrently enforcing a discernible separation between instances of disparate classes. In contrast to

conventional k-nearest neighbor (k-NN) techniques, large margin nearest neighbor stands distinguished by its capacity to enhance classifier efficacy through the acquisition of an optimal metric space. The amalgamation of large margin nearest neighbor with machine learning classifiers finds primary utility within the feature engineering phase of classifiers, wherein metric learning serves to further refine feature representation. As delineated in the associated illustration, the integration of large margin nearest neighbor with a classifier typically adheres to the ensuing steps:

(1) Preprocessing of samples: This entails the application of procedures such as data imputation and normalization to the sampled data.

(2) Acquisition of metric space: Leveraging the preprocessed samples as input features and employing classification labels as metric learning labels, the algorithm endeavors to acquire the spatial matrix M within the Mahalanobis metric space. Subsequently, this matrix is utilized to effectuate spatial transformation upon the samples.

(3) Deployment of the classifier: Traditional classifiers, including but not limited to k-NN, support vector machines (SVM), and decision trees, are invoked for the classification of features within the newly configured metric space.

## Preprocessing

In this study, we use Boolean values to encode the questionnaire data. When a patient has a certain symptom, it is marked as 1, and when there is no symptom, it is marked as 0. The data underwent preprocessing procedures, including standardization, normalization, and imputation of missing values. Due to the limited sample size, missing data was manually imputed in this experiment. Subsequently, the preprocessed data was fed into the large margin nearest neighbor algorithm (*Li & Tian, 2018*) to train the large margin nearest neighbor model, optimizing the input features and partitioning the dataset. The collected data is randomly partitioned into training and testing sets at a ratio of 7:3, with 70% of the data being used for training and 30% for testing. The optimized features were then input into a machine learning classifier for model training. Finally, the performance of the best model is evaluated using the test set. The flow chart is shown in Fig. 1.

The standardized equation are presented as Eqs. (1) and (2):

$$\sigma = \sqrt{\frac{1}{N} \sum_{i=1}^{N} (x_i - \mu_i)^2} \tag{1}$$

$$z = \frac{x - \mu}{\sigma} \tag{2}$$

where, z is the Z-zero standardized output result, $\mu$ is the average value of the overall data, $\sigma$ Is the standard deviation of the overall data.

Pre-treatment flow Fig. 2 is as follows. This article employs distinct techniques for addressing missing data through the implementation of diverse filling methods. For numerical data, such as height and weight, the mean imputation technique is employed in this study. In the case of absent Boolean data, such as questionnaire responses, this research employs the zero imputation approach. As the dataset analysed in this article comprises

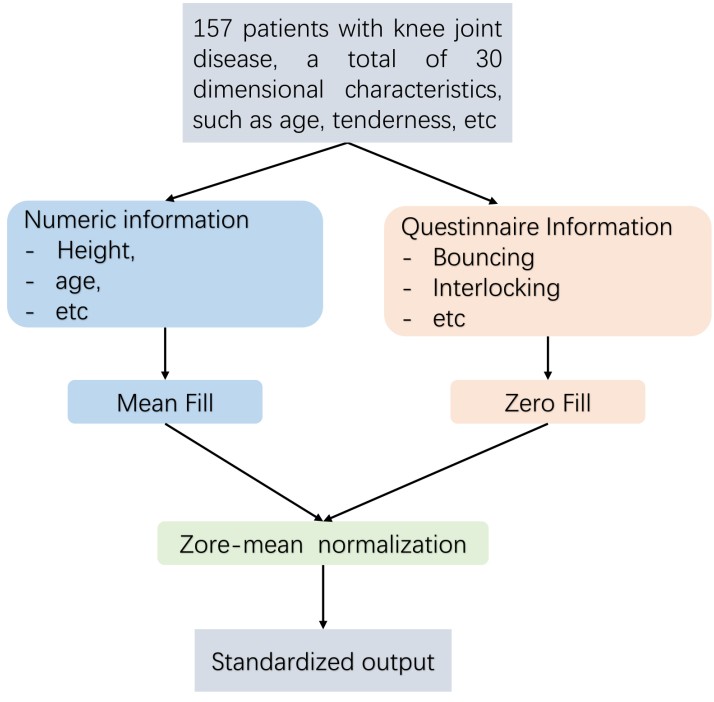

**Figure 2** Data pre-processing flow.

continuous numerical information and Boolean questionnaire data, it is necessary to eliminate any dimensional bias between diverse types of indicators. Furthermore, since distance serves as a means of measuring similarity in subsequent experiments, Z-zero normalization is implemented to standardize distinct data types within the interval of $[-1, 1]$.

## Large margin nearest neighbor models

To maximize the utilization of patient information, the feature selection step that is commonly employed in conventional modelling approaches has been omitted in the modeling process of this study. Simultaneously, due to the distinctiveness of the indicator information of various patients, the machine learning modelling process may encounter the issue of a considerable intra-class spacing, leading to ineffective classification plane division and reduced accuracy when utilizing distance-based similarity measurements. In light of the aforementioned issues, this study integrates the large margin nearest neighbor algorithm, as depicted in Fig. 3. Following the data preprocessing stage, which aims to minimize the distance between similar samples and maximize the distance between diverse sample types, the influence of sample distinctiveness on the classifier is reduced, thus optimizing the prediction outcomes.

Use a large margin nearest neighbor learning metric matrix to optimize distance relationships. This algorithm first transforms the metric in Euclidean space into a Mahalanobis space. Afterwards, learn the corresponding transformation matrix in Mahalanobis space. And the two metric calculations are as follows Eqs. (3) and (4):

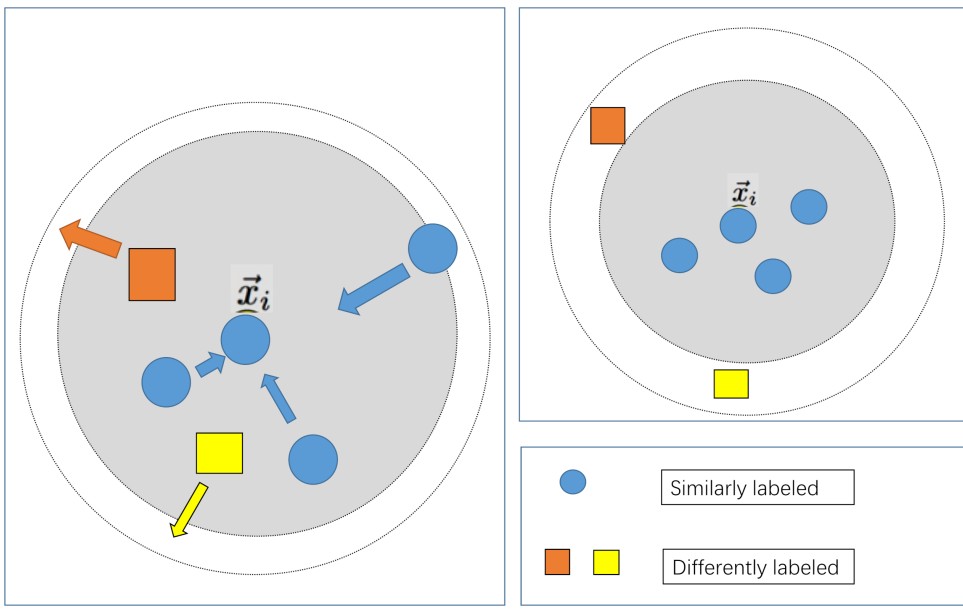

**Figure 3  Schematic diagram of large margin nearest neighbor.**

$$D(\overrightarrow{xi}, \overrightarrow{xj}) = \sqrt{\sum_{k=1}^{n}(\overrightarrow{xik} - \overrightarrow{xjk})^2} \tag{3}$$

$$D_M(\overrightarrow{xi}, \overrightarrow{xj}) = \sqrt{(\overrightarrow{xi} - \overrightarrow{xj})^T \sum{}^{-1}(\overrightarrow{xi} - \overrightarrow{xj})} \tag{4}$$

where, $D(\overrightarrow{xi}, \overrightarrow{xj})$ is the Euclidean distance calculation formula, $D_M(\overrightarrow{xi}, \overrightarrow{xj})$ represents the Mahalanobis distance, $\overrightarrow{xi}$ and $\overrightarrow{xj}$ is the sample characteristics. K is the feature dimension. Here, $\sum$ denotes the covariance matrix of multidimensional random variables, serving as weight coefficients, thereby transforming the fixed unit distance of Euclidean distance into an unequal unit distance.

It can be clearly seen from the above formula that Euclidean distance describes the straight-line distance between two points in n-dimensional space, while Mahalanobis distance represents the degree of deviation between points and distributions. Due to the introduction of covariance matrix, Mahalanobis distance can better represent the global distribution of samples.

In order to project the Euclidean space onto the Mahalanobis space, we need to find the linear transformation matrix L, so that the distance in the transformed feature space is equivalent to the Mahalanobis distance. According to the above Euclidean distance and Mahalanobis distance calculation formulas, L can be calculated by the inverse square root of the covariance matrix.

Compared with traditional Euclidean distance measures, Mahalanobis distance shows superior performance in identifying similar types of samples, thereby promoting more cohesive sample clustering. On this basis, large margin nearest neighbor continues to learn the corresponding transformation matrix in the Mahalanobis space, iteratively updating and refining the metric space through the loss function shown in the following formula. Convert the Euclidean metric to the Mahalanobis metric as the initial transformation matrix for the large margin nearest neighbor algorithm, then calculate the loss and iterate according to the following as follows Eq. (5) (*Weinberger & Lawrence, 2009*) to learn a more accurate transformation matrix (*Bellet, Habrard & Sebban, 2013*):

$$\varepsilon(L) = \sum_{ij} \mu_{ij} ||L(\vec{xi} - \vec{xj})||^2 + c \sum_{ijl} \mu_{ij}(1 - y_{il})\left[1 + L(||\vec{xi}) - \vec{xj}||^2 - ||L(\vec{xi} - \vec{xl})2\right]_+ \quad (5)$$

where, $\vec{xi}$ represents an arbitrary input sample, $\vec{xl}$ serves as a surrogate (located in close proximity but not belonging to the same class), and $\vec{xj}$ denotes a sample that belongs to the same class as $\vec{xi}$ and is in close proximity to it. If $\vec{xi}$ and $\vec{xj}$ are neighboring samples of the same class, then $\mu_{ij}$ takes the value 1; otherwise, it is 0. Similarly, if $\vec{xi}$ and $\vec{xl}$ are neighboring samples of different classes, then $y_{il}$ is set to 1; otherwise, it is 0. + ensures that only positive values are considered, and it is set to 0 if the value is less than 0. C represents a weighting coefficient. Given the absence of relevant prior knowledge, this study initializes the labels using the euclidean distance for measurement purpoese. $||L(\vec{xi} - \vec{xj})||^2$ representing transformation matrix obtained under the Mahalanobis metric.

The first term only penalizes large distances between inputs and target neighbors. The second item is used to punish the problem of maximum class spacing, and c needs to be greater than zero. $\varepsilon(L)$ is a distance function trained to replace the traditional fixed distance measurement.

The introduction of the second term in the loss function engenders the notion of margin. Specifically, the hinge loss incurred by each input feature vector is influenced by samples bearing distinct labels and confined within a prescribed threshold. Consequently, a principal function of the cost function (the second term) entails augmenting the disparity between samples of dissimilar labels, thereby delineating a significant margin, whereas the first term (the identical term) serves to diminish the inter-sample distance within identical label sets.

Thus, the resultant sample feature space manifests diminished intra-class spacing juxtaposed with amplified inter-class spacing. In the present discourse, preprocessed features serve as input for metric learning, while category labels assume roles as metric learning labels during training. Subsequent to the application of the metric matrix M, the original data undergoes transformation, thereby yielding data situated within a novel metric space.

According to the metric transformation matrix M trained by the large margin nearest neighbor model mentioned above, the Euclidean distance from the original sample can be projected onto a new Mahalanobis metric space. Afterwards, the samples in the new metric space are used as inputs for machine learning classifiers and fed into different classifiers.

## Classifier

This study involved preprocessing and employed large margin nearest neighbor for the treatment of original features. The processed features retained their original dimensions. Subsequently, the processed sample features were partitioned into training and testing sets. Various machine learning classifiers were then trained using the training set. Building upon the foundation laid by *Wang et al. (2021)*, this research utilized support vector machines, logistic regression (LR), Adaboost, multilayer perceptron (MLP) and Random Forest (RFF) algorithms as predictive classification models to assess the efficacy of the proposed methodology.

## EXPERIMENT

The Ethics Committee of Hefei Institutes of Physical Science, Chinese Academy of Sciences granted approval for the study (SWYX-Y-2021-13). The participant received detailed information about the test procedures and provided their informed consent to partake in the experiment. Throughout the experiment, the patients were initially provided with foundational knowledge of the knee *via* the knee joint education system. Following this, the patients completed a systematic diagnostic questionnaire with the assistance of medical professionals.

Between June 2019 and June 2020, participants were recruited from the Department of Orthopedics at the First Affiliated Hospital of the University of Science and Technology of China. A questionnaire was then administered randomly to patients with knee joint conditions. The disease diagnosis was verified by a group of expert physicians. The patient completed the questionnaire based on current symptoms and retrospective feedback, and subsequently provided their signature on the informed consent form.

Informed consent was obtained from all individuals participating in the study. The present study adheres to the principles set forth in the Declaration of Helsinki and has been granted approval by the Ethics Committee of the Chinese Academy of Sciences' Hefei Materials Research Institute. The current experiment involved the administration of a sampling questionnaire to 157 patients with knee joint diseases. During the experiment, a team of three medical professionals assisted with the completion of the questionnaire to enhance the precision of symptom information. Furthermore, the team confirmed the patients' disease classification based on hospital examinations, such as fusion imaging and laboratory analyses. The detailed content and related analysis of the questionnaire were published in *Wang et al. (2021)*.

This study employs various machine learning classification algorithms to categorize diverse combinations of symptom features. The evaluation of classifier performance is predominantly conducted through six assessment parameters: accuracy, precision, recall, F1 score, ROC curve, and AUC. The purpose is to identify the most suitable classifier based on effectiveness. The meanings of the six evaluation parameters and their abbreviations are elucidated as follows:

(1) True positive (TP): The number of samples that are truly positive and predicted as True Positive.

(2) False positive (FP): The samples that are actually negative but are incorrectly predicted as False Positive.

(3) True negative (TN): The instances where the samples are genuinely negative and predicted negativeas as True Negative.

(4) False negative (FN): The cases where the samples are truly positive, but the prediction erroneously indicates a negative outcome as false negative.

The relevant evaluation parameters are defined as follows:

(1) Accuracy (ACC): The proportion of correctly classified instances among the total samples, The calculation equation are as follows Eq. (6):

$$ACC = (TP + TN)/(TP + TN + FP + FN) \tag{6}$$

(2) Precision (P): The ratio of true positive predictions to the total instances predicted as positive, providing an indication of the classifier's ability to correctly identify positive cases, The calculation equation are as follows Eq. (7):

$$P = TP/(TP + FP) \tag{7}$$

(3) Recall (R): The ratio of true positive predictions to the total actual positive instances, reflecting the classifier's capacity to capture all positive cases, The calculation equation are as follows Eq. (8):

$$R = TP/(TP + FN) \tag{8}$$

(4) F1 Score (F1): The harmonic mean of precision and recall, offering a balanced measure that considers both false positives and false negatives, The calculation equation are as follows Eq. (9):

$$F1 = (2 \times P \times R)/(P + R) \tag{9}$$

(5) ROC curve: A graphical representation illustrating the trade-off between true positive rate (sensitivity) and false positive rate (1-specificity) across different classifier thresholds.

(6) AUC: The area under the ROC curve, providing a single scalar value to assess the classifier's overall discriminatory ability, with higher values indicating better performance.

Concerning feature selection, this study incorporated all information from the questionnaire, encompassing patient characteristics such as gender, age, height, weight, knee joint workload, affected knee position, activity restrictions, injury status, bursting sensation, locking, instability, knee dislocation, patellar dislocation, stiffness, swelling, pain, and tenderness.

This experiment was conducted on Windows, using Python 3.9 as the compilation environment, as long as the sklearn and metric_learn libraries were used to build the code.

This article first imported and manually preprocessed the collected data of 157 cases. Import each user as a row and each piece of information as a column into the code model. For the missing data, the questionnaire information was filled with zero padding. For numerical data such as height and weight, this article used mean padding to calculate the mean of each complete data item, and then assigned the mean of each item to these missing data.These processes were manually completed in Excel for this article. Finally,

**Table 2  Test set experimental results.**

|  | Accuracy | Recall | Precision | F1 |
|---|---|---|---|---|
| SVM+preprocessing | 0.81 | 0.79 | 0.88 | 0.83 |
| RRF+preprocessing | 0.71 | 0.74 | 0.68 | 0.71 |
| AdaBoost+preprocessing | 0.75 | 0.76 | 0.76 | 0.76 |
| MLP+preprocessing | 0.81 | 0.81 | 0.84 | 0.82 |
| LR+preprocessing | 0.79 | 0.78 | 0.84 | 0.81 |
| SVM+preprocessing+LMNN | 0.81 | 0.79 | 0.88 | 0.83 |
| LR+preprocessing+LMNN | 0.81 | 0.79 | 0.88 | 0.83 |

normalization is adopted to normalize the information at different scales into the $[-1,1]$ space, ensuring the consistency of data morphology.

After completing data preprocessing, we feed all the data into the large margin nearest neighbor algorithm, treating each user as a sample, the original features as feature tensors, and where they are ill as labels to train the large margin nearest neighbor model In terms of large margin nearest neighbor parameter settings, in the optimal case, large margin nearest neighbor were employed for binary classification with a learning rate set at $1e-4$ and $k = 2$ due to preprocessing, the knee joint data is linearly separated in the feature space. Other parameters use the large margin nearest neighbor algorithm's build in parameters in the metric learning library under Python.

The data processed by large margin nearest neighbor is divided into training and testing sets using a random function in a 7:3 ratio. Afterwards, the training set is fed into different classifiers for training, and the performance of different classifiers is evaluated using the test set. In the comparative experiment, this article processes the large margin nearest neighbor part and directly sends the raw data to different classifiers after processing according to the above steps. In configuring classifier parameters, regularized logistic regression utilized the liblinear kernel with a maximum iteration setting of 40. In contrast, SVM employed a linear kernel, and the determination of the maximum iteration relied on a heuristic method for convergence. Adaboost had a maximum iteration setting of 10. The Random Forest count in random forest was set to 3, with a maximum feature number of 0.5. The hidden layer size of the Multilayer Perceptron was (50, 50), with an initial learning rate of $1e-3$, adaptive adjustment during iteration, a maximum iteration setting of 5,000, 'identity' as the activation function, 'sgd' as the solver, and alpha set to 5. All other parameters used default values from relevant classifier functions in the sklearn library.

In the comparative experimental setup, preprocessing and classifier processes were retained, maintaining consistency in experimental parameters to evaluate the impact of large margin nearest neighbor on knee joint prediction classification results.

In this study, various bechmark machine learning models were assessed on a proprietary dataset, and comparisons were drawn with enhanced models. The specific outcomes are delineated in the ensuing Table 2, as depicted below.

Table 2 illustrates that the model developed in this study achieved optimal performance by utilizing all 30 features from the experiment. In the comparative experiment, we used commonly used machine learning classifiers in disease prediction relevant work as

baseline models for comparative experiments (*Zhang et al., 2023*). Notably, the LR classifier exhibited a significant performance improvement, while SVM showed no substantial changes. This discrepancy can be attributed to SVM relying on a small set of salient samples to determine the classification plane, whereas LR utilizes all samples, making it more sensitive to the impact of specific samples. This finding suggests that the integration of metric learning effectively addresses challenges related to inter-class distances, resulting in a considerable enhancement in F1-score. Consequently, this model is deemed more suitable for addressing scenarios with a large sample size and strong specificity, such as disease prediction.

Figures 4A and 4B depict the experimental results of SVM and LR. Figures 4C and 4D present the experimental outcomes of this research. Under the SVM classifier, the area under the curve (AUC) increased from 0.78 to 0.82, and for LR, it improved from 0.82 to 0.84. Despite some intersecting trends observed in the curves, the results of this study significantly outperformed the previous research findings. The ROC curves offer better insights into the impact of large margin nearest neighbor on classification outcomes, enhancing the precision of each sample's determination probability. This indicates that large margin nearest neighbor optimizes the classifier's hyperplane, improves inter-class distances, enhances the classification accuracy of samples with lower specificity, and reduces the influence of sample specificity on classification results. In future research, it is necessary to validate the effectiveness of the large margin nearest neighbor model using a larger dataset and to incorporate metric learning into other machine learning models related to disease prediction, in order to evaluate its impact.

## CONCLUSIONS

In this study,we employ metric learning to replace the conventional feature selection steps in traditional machine learning modelling techniques for knee joint diseases, which are known to be highly specific and complex. By incorporating more input information, we can further decrease the dependence on expert evaluation and screening of input information or the opaque nature of machine learning feature selection. This approach serves as a means to mitigate the influence of human-driven feature selection on the predictive model, enhance its specificity and accuracy, and render it more suitable for modeling medical data. This approach presents a novel proposition for utilizing machine learning techniques in the domain of medical prediction by offering a clean and accessible dataset for model training purposes. This implies that the model can be trained to capture the underlying relationship between features and diseases more accurately, resulting in improved prediction performance compared to the previous state-of-the-art. Metric learning has great potential in processing medical questionnaire information and disease prediction. In future work, we will verify its reliability in clinical application and try to prove its role in other medical related fields.

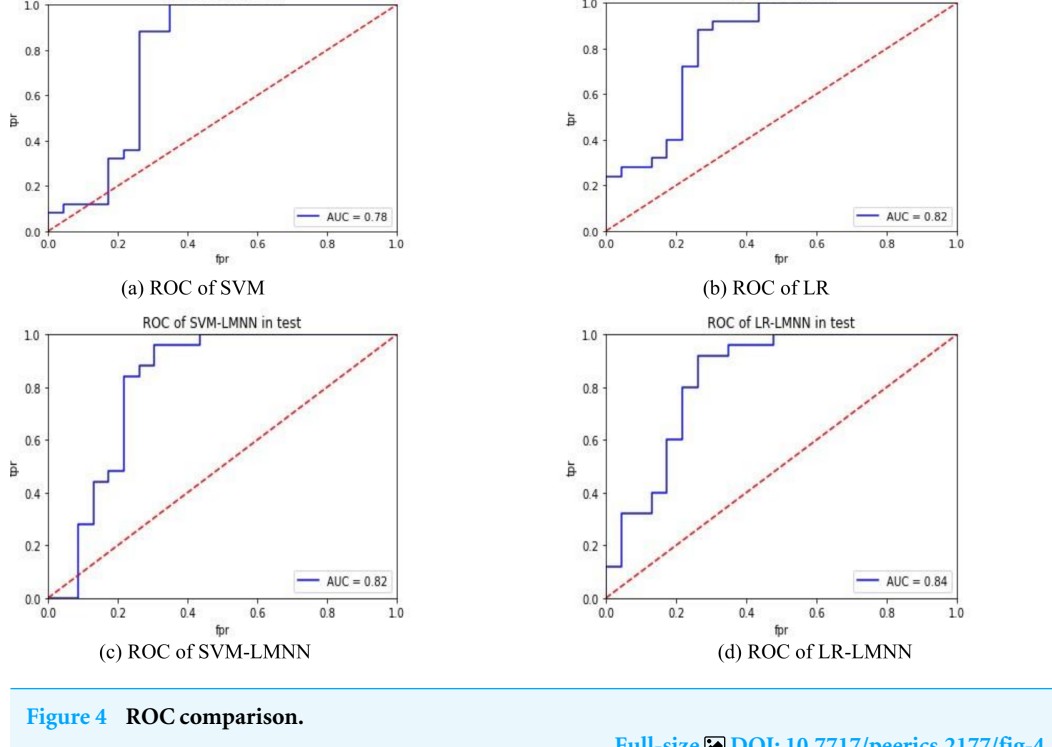

(a) ROC of SVM

(b) ROC of LR

(c) ROC of SVM-LMNN

(d) ROC of LR-LMNN

**Figure 4** ROC comparison.

### Funding

This research was supported by the Major projects of Anhui science and technology (No. 202103a07020004). The funders had no role in study design, data collection and analysis, decision to publish, or preparation of the manuscript.

### Grant Disclosures

The following grant information was disclosed by the authors:
The Major Projects of Anhui Science and Technology: No. 202103a07020004.

### Competing Interests

The authors declare there are no competing interests.

### Author Contributions

- Yu Wang performed the experiments, analyzed the data, performed the computation work, prepared figures and/or tables, authored or reviewed drafts of the article, and approved the final draft.
- Yiwei Liang analyzed the data, prepared figures and/or tables, and approved the final draft.
- Guangjun Wang performed the experiments, analyzed the data, prepared figures and/or tables, and approved the final draft.

- Tao Wang analyzed the data, prepared figures and/or tables, and approved the final draft.
- Shu Xu performed the computation work, prepared figures and/or tables, and approved the final draft.
- Xianjun Yang conceived and designed the experiments, authored or reviewed drafts of the article, and approved the final draft.
- Yining Sun conceived and designed the experiments, authored or reviewed drafts of the article, and approved the final draft.
- Zenghui Ding conceived and designed the experiments, authored or reviewed drafts of the article, and approved the final draft.

### Ethics

The following information was supplied relating to ethical approvals (i.e., approving body and any reference numbers):

The Ethics Committee of Hefei Institutes of Physical Science, Chinese Academy of Sciences granted approval for the study (SWYX-Y-2021-13).

### Data Availability

Code and data are available at GitHub and Zenodo:

https://github.com/wy1991ty/MeniscusDamagePrediction

Wang, Y. (2024). Meniscus Injury Prediction Model based on Metric Learning. Zenodo. https://doi.org/10.5281/zenodo.12622775.

### Supplemental Information

Supplemental information for this article can be found online at http://dx.doi.org/10.7717/peerj-cs.2177#supplemental-information.

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
