# Peer review of "Meniscus injury prediction model based on metric learning"

_PeerJ Computer Science, doi:10.7717/peerj-cs.2177_

## Round 0.1 · original submission · Major Revisions

Please take into account the comments of the reviewers.

**Language Note:** The review process has identified that the English language must be improved. PeerJ can provide language editing services - please contact us at [email protected] for pricing (be sure to provide your manuscript number and title). Alternatively, you should make your own arrangements to improve the language quality and provide details in your response letter. – PeerJ Staff

Reviewer 1 ·

Basic reporting

The authors need to indicate the enhancement of the accuracy percentage in the abstract.
 Some of the references and citations are missing in the text. Need to revise carefully.
 The authors should include the section “Our contribution” and need to mention the contributions and novelty of the work clearly in bullet points.
 The organization of the paper section is missing.
 A literature review is missing. Which is an important part of the research paper.
 The authors must include the literature review. There must be a table of literature review which clearly shows the existing scheme methodology, objectives, performance and measures and most importantly the limitations. Which motivates the authors to work on this topic.
 The authors need to mention and explain deeply about features, labels means square error etc. Haven’t seen any details of the technical part of the work.
 The authors need to do benchmarking to show the best performance of the proposed scheme.
 The authors should update the references.

Experimental design

Presented well and cleared.

Validity of the findings

Cleared and Shown.

Cite this review as

·

Basic reporting

English language. Text contains various language inaccuracies, for example the same terms written in capital letters, or in lower case letters. Often there is no blank space after closing bracket.

Abstract and introduction clearly state the problem presented in the text.

The general structure of the text is good and it is withing the PeerJ standards. However, the text is extremely short for a journal article.

Figures in Fig. 4 are blurry. It is better to use vector graphics to include figures in the text.

Raw data. The authors share the source code written in Python as part of the research. It makes no sense to share source code for an English language journal with comments in Chinese. Although the lack of English documentation of the source code, it is obvious from the program that the research is in an initial state, and maybe authors did not complete it yet.

Experimental design

The research topic is within the scope of the journal. However, it is not obvious that it is original. The usage of metric learning is a popular and a well-known technique. For example, the process given in Fig. 1 is well known textbook fact.

The application of the research is well defined. However, the enhancement, presented by authors is not obviously original. It is a known method.

Technical standard of the research is hard to be evaluated, because the shared source code shows an implementation is some kind of initial state.

Methods described are not in sufficient detail & information to replicate. The text does not contain an algorithm, or detailed description of the method.

Validity of the findings

It is hard to evaluate impact and novelty in the presented work, because it addresses the application of known technique. See for example:
Dewei Li, Yingjie Tian, "Survey and experimental study on metric learning methods", Neural Networks 105, 447-462, 2018.

The underlying data is stored in a relatively short electronic table, that can hardly make a statistical sense.

Additional comments

The result reported in this text looks like in initial state, which is not yet completed enough to be published as a journal paper.

Cite this review as

---

## Round 0.2 · Major Revisions

Please accurately check the comments of the reviewers.

Reviewer 1 ·

Basic reporting

The authors incorporated the comments accordingly.

Experimental design

N/A

Validity of the findings

N/A

Additional comments

N/A

Cite this review as

·

Basic reporting

## English language

In many places there is no space character between a preceding word and phrase in brackets, for example:
> artificial intelligence(AI)
There are too many abbreviations used in the text.

In a computer science text, do not refer to a mathematical expression as *formula*. Call them, for example:
> see equation (1)

## Intro, background and references

Introduction focused too much on the application. The main focus should be on the computer science aspect of the proposition.

## Text structure

Contribution description must be part of introduction and conclusion. There is no need for a separate small section.

## Figures and tables

Figures and tables are clear and readable.

## Raw data

Authors provide the source code of their experiment written in Python.

Experimental design

## Originality of the research and scope of the journal

The topic of the proposed text is within the scope of the journal.

## Research questions definition

Two parts of the research are clear: the medical application and the incorporation of metric learning. There is a huge gap in this text in the definition of exactly what metrics the metric learning is used for and how exactly the result of it is incorporated in to machine learning phase. This is exactly where the should be focused.

## Technical and ethical standards of the research

Te representation of the real part of research is unclear and insufficient.

The mathematical expressions in the text are either incorrectly written, or are unclear, since authors do not describe their contents. Please note that the symbol * does not mean multiplication in scientific mathematical expressions.

## Description detail sufficiency to replicate

Description detail is not sufficient to replicate.

Validity of the findings

## Impact and novelty

Both impact and novelty are questionable. Medical application is important, however it does not mean the automatically any suggestion how to process this type of data is novel from computer science point of view. Metric learning is a well known technique, see for example:
Bellet, A.; Habrard, A.; Sebban, M. (2013). "A Survey on Metric Learning for Feature Vectors and Structured Data".

## Data provided robustness and statistical control

The provided dataset is in a for of electronic table with binary values with 143 rows and 15 columns. The authors did not explain the meaning of this binary matrix. We cannot approve any statistical control in this case.

Additional comments

As I have mentioned in my first review, this proposition is in an initial stage. The incorporation of the metrics learning into the machine learning phase is unclear, and exactly this should be the real contribution of this text.

Cite this review as

---

## Round 0.3 · Minor Revisions

Please do the final minor revisions requested by reviewer 3

Reviewer 1 ·

Basic reporting

The authors addressed all the comments accordingly.

Experimental design

N/A

Validity of the findings

N/A

Additional comments

N/A

Cite this review as

Reviewer 3 ·

Basic reporting

See below.

Experimental design

See below.

Validity of the findings

See below.

Additional comments

After analyzing the document sent and the author's response letter, I recommend some small changes to the authors:
• Table 1 caption has “table 1” twice
• “Table 2 illustrates that the model developed in this research achieved optimal performance by utilizing all 30 features in the experimentation and was compared with previous relevant results.”
Suggestion: In the table, each row should have the reference to the paper from where these results are gathered (if you didn’t implement and run your dataset in the models, of course)
• In Conclusions, the initial 2 sentences are claiming the same… review and re-write. Some future directions would also be a plus.

Cite this review as

---

## Round 0.4 · Major Revisions

As the comments of the previous review were not fully addressed. I recommend the authors to fix the comments below:
1. Some spacing problems between abbreviations and the preceding text are still present. For instance, "Machine Learning(ML)" should have a space like "Machine Learning (ML)."
2. The introduction provides more focus on the computer science aspect, particularly on the integration of metric learning. However, it could emphasize even more why this is innovative within computer science.
3. While the incorporation of metric learning is explained in greater detail, it should still clearly define the specific metrics used and how these are incorporated into the machine learning phase.
4. The methodology is more detailed than before, but explicit details on how the experiments can be replicated are still somewhat lacking.
5. The work now emphasizes the novelty in computer science. However, while the potential impact is more clearly explained, the novelty in computer science remains questionable due to the already established applications of metric learning.
6. The integration of metric learning is described better, but further clarity is needed on how it exactly benefits classification in this specific application.

Reviewer 3 ·

Basic reporting

NA

Experimental design

NA

Validity of the findings

NA

Additional comments

The authors fulfilled all requests.

Cite this review as

---

## Round 0.5 · accepted · Accept

Dear authors,.

My previous feedback was addressed, Congratulations!

Reviewer 3 ·

Basic reporting

NA

Experimental design

NA

Validity of the findings

NA

Additional comments

Like before, the manuscript is ready to be accepted. All my previous concerns were addressed.

Cite this review as